# The Impact of Software Used and the Type of Target Protein on Molecular Docking Accuracy

**DOI:** 10.3390/molecules27249041

**Published:** 2022-12-18

**Authors:** Larisa Ivanova, Mati Karelson

**Affiliations:** Institute of Chemistry, University of Tartu, Ravila 14a, 50411 Tartu, Estonia

**Keywords:** molecular docking, high-throughput virtual screening, binding energy, ligand efficiency, neurodegenerative diseases

## Abstract

The modern development of computer technology and different in silico methods have had an increasing impact on the discovery and development of new drugs. Different molecular docking techniques most widely used in silico methods in drug discovery. Currently, the time and financial costs for the initial hit identification can be significantly reduced due to the ability to perform high-throughput virtual screening of large compound libraries in a short time. However, the selection of potential hit compounds still remains more of a random process, because there is still no consensus on what the binding energy and ligand efficiency (LE) of a potentially active compound should be. In the best cases, only 20–30% of compounds identified by molecular docking are active in biological tests. In this work, we evaluated the impact of the docking software used as well as the type of the target protein on the molecular docking results and their accuracy using an example of the three most popular programs and five target proteins related to neurodegenerative diseases. In addition, we attempted to determine the “reliable range” of the binding energy and LE that would allow selecting compounds with biological activity in the desired concentration range.

## 1. Introduction

The identification of drug candidates is one of the most laborious and difficult steps in the development of new drugs [1,2]. However, the development of different computer-aided drug design (CADD) techniques, as well as the increase in the number of known three-dimensional (3D) structures of protein targets in recent decades, has made it possible to reduce both time and financial costs for the search and development of new drugs [3]. The main CADD tools are quantitative structure–activity relationship (QSAR), pharmacophore modeling, homology modeling, molecular docking and molecular dynamics, and high-throughput virtual screening (HTVS) [3,4]. Molecular docking and HTVS have gained considerable importance in CADD, with the following major objectives: (I) the possibility of rapidly screening large compound libraries, (II) ranking compounds by calculated binding free energy values, (III) prioritizing binding modes to the studied target and (IV) assisting the interpretation of experimental observations [5,6]. Over the past two decades, about 60 different computer programs for molecular docking have been developed [7]. The main goal of a molecular docking program is the prediction of a possible binding mode of a small-molecule ligand at the active site of the target protein. Thereafter, based on the predicted pose, the docking program estimates the binding affinity (Gibbs free energy of the binding or docking score) [8,9]. The predicted binding free energy is one of the main parameters that allow for the initial separation of possible binding ligands from molecules that are unlikely to bind a target [9,10]. However, such initial discrimination may leave some active compounds out. Another important criterion for the initial filtering of HTVS results and for the further selection of potential candidates for experimental testing is ligand efficiency (LE)— “a useful metric for lead selection” [11,12]. The most common values for the selection of potential candidates that are currently accepted in drug design are values less than −6.0 kcal/mol (in some cases, lower than −8.0 kcal/mol) for binding free energy [13] and values greater than 0.3 kcal/mol per heavy atom (i.e., a non-hydrogen atom) for LE [14,15]. Both parameters are routinely used in drug design [16,17,18]. However, there is still no consensus on the range that these parameters should fall within for biologically active compounds [19]. According to the published data, typical hit rates from experimental screening can range between 0.01% and 0.14%, whereas HTVS typically gives hit rates between 1% and 40% [20,21]. One of the main reasons for the still relatively low efficiency of HTVS and molecular docking may be due to simplifications and assumptions in the scoring functions that serve to increase the calculation speed. The main simplifications are (I) neglecting water explicitly as a solvent by most docking programs, (II) neglecting conformational states of the protein, which is often treated as a rigid body, and (III) the use of simple potential energy functions, such as force fields or statistical potentials [18]. These simplifications of protein–ligand interaction modeling are manifested by inaccuracies in pose ranking and poor performance in the prediction of binding free energies [10]. The available molecular docking programs apply different types of force fields, scoring functions and search algorithms, and each of these can affect the predictive power of virtual screening and molecular docking. By evaluating the accuracy of docking pose predictions and the virtual screening of six different molecular docking programs, Cross et al. showed that modifying the basic parameters in the molecular docking program has a significant effect on docking and virtual screening results [22]. In addition, the structure, functions, and features of the target protein can also have a great influence on the results of molecular docking. Wang et al. demonstrated that the predictive power of each of the ten studied docking programs is quite different for different protein families [23]. This dependence on the type of target protein is obviously due to the different types of interactions playing a crucial role in ligand binding. For non-covalent binding, the protein–ligand interactions include ionic interactions, hydrogen bonds and Van der Waals (VdW) interactions (including dispersion, polar and induced interactions). Theoretically, these different types of interactions can be estimated in the most accurate way by using quantum mechanics. However, despite enormous progress in computational power, this approach is still prohibitively expensive for HTVS purposes [24]. Although currently used force fields and scoring functions are, in general, well parametrized, properly accounting for polarization effects and detailed proton affinity estimations are still lacking [18]. The efficiency of HTVS or molecular docking can be improved by visually inspecting the binding poses of compounds with the best binding energy or LE [25]. However, this is often subjective and, in the case of large-scale HTVS, can be complicated due to a large number of hit compounds and docked poses [9]. Moreover, it is worth noting that the accuracies of the binding free energy predictions of most available docking programs have a standard deviation of about 2–3 kcal/mol, which is not sufficient to provide a confident ranking of compounds [26,27]. Therefore, to date, the main goal of molecular docking remains the “enrichment” of the original library with compounds that show strong binding upon testing. Thus, proper prediction of the binding energy and LE, as well as the related software selection, remain the first and most important features influencing the efficiency and success of the early stages of drug development.

The main goal of this work was to evaluate the impact of the docking software as well as the type of the target protein on the molecular docking results and its accuracy using an example of the three most popular programs [3] (AutoDock 4.2.6 [28], AutoDock Vina 1.1.2 [29] and GLIDE [30,31,32]) and five target proteins related to neurodegenerative diseases. In addition, we attempted to identify a “reliable range” of binding energy and LE and to evaluate its dependence on the software used.

### 1.1. Target Proteins

With an improvement in the quality of life and medical services, life expectancy is also increasing, which in turn leads to an aging population and the spread of age-related diseases, particularly neurodegenerative diseases. Many organizations and individual laboratories are working on the development of treatment for neurodegenerative diseases, but there is still no effective therapy. The main obstacles are the complexity of the brain and its working mechanisms, insufficient availability of resources and ethical limitations. However, advances in modern computer technology have allowed for avoiding these limitations. Currently, enough protein targets have been found to be directly or indirectly associated with neurodegenerative diseases [33]. Such a variety of potential targets arises from the fact that most neurodegenerative diseases are characterized by multiple disorders. In our work, five proteins related to Alzheimer’s (AD) and Parkinson’s (PD) diseases were selected as target proteins for molecular docking (Table 1).

Each of the selected proteins belongs to a different class and has its own features and characteristics, which can have a significant impact on the docking results. The most distinct feature of acetylcholinesterase (AChE) is its exceptional active site structure (Figure 1A). The AChE active site represents a deep and narrow gorge about 20 Å long, which penetrates more than halfway into the enzyme and widens out close to its base [46]. This structure of the active site may be the reason why, during molecular docking, active compounds with a relatively large molecular size are automatically excluded. It can be assumed that, in the case of AChE, the results of the virtual screening largely depend on the search algorithm used to treat ligand flexibility and the accuracy of the scoring function. In the case of beta-site amyloid precursor protein cleaving enzyme 1 (BACE1) (Figure 1B) and serotonin transporter (SERT) (Figure 1C), the active site has an open conformational state, which directly affects interactions with the ligand, and, accordingly, should be considered during the docking procedure [43,47]. Another feature of SERT is the presence of an additional (allosteric) binding site (Figure 1D), the occupancy of which sterically hinders ligand unbinding from the central site [43]. Thus, it is reasonable to assume that small-molecule ligands should be docked at both active sites, which increases the amount of work and complicates the selection of potential candidates. In the development of new inhibitors of glycogen synthase kinase 3 beta (GSK3β) (Figure 1E) and tropomyosin receptor kinase A (TrkA) (Figure 1F), the main challenge is to secure the selectivity inhibitors due to the multiple roles of GSK3β and TrkA in the regulation of cellular processes [48,49]. A molecular dynamics simulation study and a thorough analysis of the binding poses of known GSK3β inhibitors by Arfeen et al. showed that strong hydrogen bonding with Lys85 has a crucial impact on the selectivity of most GSK3β inhibitors [50]. In addition to specific hydrogen bonding, as in the case of GSK3β, hydrophobic interactions in and around the binding site of the adenine ring of ATP are also important for TrkA inhibitors [51]. Thus, the force field and the scoring function can have a strong influence on the results of virtual screening during the development of new inhibitors of GSK3β and TrkA.

### 1.2. Molecular Docking Software

#### 1.2.1. AutoDock 4.2.6

The AutoDock program was developed in 1989 by the Scripps Research Institute and remains one of the most cited molecular docking software. For instance, the paper by Morris et al. [28], published in 2009 and reporting on the 4th version of AutoDock, as of October 2022, has been cited more than 15.5 thousand times. AutoDock is a free available molecular docking program based on a semiempirical free energy force field and has a variety of search algorithms, including a Monte Carlo Simulated Annealing algorithm, a Genetic Algorithm (GA) and a rapid hybrid local search GA, also known as the Lamarckian Genetic Algorithm (LGA) [26,52,53]. The major advances of the semiempirical free energy force field are the use of an improved thermodynamic model of the ligand–protein binding process and a full desolvation model, including terms for all atom types. Moreover, this type of force field uses an improved model of directionality in hydrogen bonds that allows for predicting the proper alignment of groups with multiple hydrogen bonds [26]. In the present work, the LGA was used as a search algorithm because it gives more reliable and efficient results and makes it possible to operate with ligands with a large number of degrees of freedom [53]. However, it is worth noting that a number of rotatable bonds in the ligand greater than six may cause inaccurate results [54]. One of the disadvantages of AutoDock is the time required to prepare input files and the relatively slow speed of calculations in the case of docking ligands one by one. In our experience, docking a single compound with a molecular weight of 350 to 500 Da can take up to 15 min with 25 AutoDock runs, which is not appropriate for HTVS. However, Scripps Research Institute has now developed an additional graphical interface, Raccoon, that automatically carries out some of the most common HTVS preparation steps and generates scripts for running HTVS [55]. In the present study, Raccoon v.1.0f [55] was used for the preparation of input files and for the generation of HTVS scripts.

#### 1.2.2. AutoDock Vina 1.1.2

AutoDock Vina 1.1.2 (hereinafter referred to as Vina 1.1.2) is one of the open-source docking engines of the AutoDock Suite that was developed and initially released in 2010 by Dr. Oleg Trott in the Molecular Graphics Lab at The Scripps Research Institute [29]. Like AutoDock 4, Vina is also one of the most cited (over 21,000 citations) software used for molecular docking. Depending on the task and its complexity, Vina can run up to 100 times faster than AutoDock 4.2.6 [29]. The improved efficiency of calculations is ensured using a Monte Carlo iterated search algorithm combined with the Broyden–Fletcher–Goldfarb–Shanno (BFGS) [56] gradient-based optimizer that in turn leads to better docking results with fewer scoring function evaluations. The Vina scoring function is based on the combination of advantages of knowledge-based potentials and empirical scoring functions. The program extracts empirical information from both the conformational preferences of receptor–ligand complexes and experimental affinity measurements [29]. For the calibration of both the AutoDock 4.2.6 and Vina programs, the same test set of 30 structurally known protein–ligand complexes was used [28,29]. However, it has been noted that Vina demonstrates significantly better average accuracy for binding mode predictions [29,57]. Moreover, according to the results of independent tests, Vina was recognized as “a strong competitor against the other programs, and at the top of the pack in many cases” [58]. In the case of Vina, it is not needed to run separate docking or virtual screening calculations and to generate docking parameter and grid map files. Only PDBQT files are required, which can be easily generated using the Raccoon graphical interface [55]. Due to this, Vina can be easily adapted for HTVS of large libraries of compounds in batch mode [59].

#### 1.2.3. Glide Docking

Glide is a commercially available program implemented in the Schrödinger Maestro software. The program was released in 2004 as “a new approach for rapid, accurate docking and scoring” [30,32]. Along with AutoDock software, Glide is a frequently used molecular docking program [3] that uses an empirical scoring function in its work. The use of the OPLS (Optimized Potentials for Liquid Simulations) force field allows for comprehensive orientational, conformational and positional ligand binding pose calculations followed by subsequent conformational refinement using the Monte Carlo search algorithm [3,30,31,32]. According to the published data, Glide binding pose prediction accuracy is approximately twice as high as the accuracy of other molecular docking programs (GOLD and FlexX) [30]. The main advantage of this molecular docking software is its intuitive protocol, making it easy to use for beginners, and its minimal requirements and time for preparing input files. However, the commercial distribution of this software may limit its applicability. The virtual screening of a large number of compounds can be performed using a specially designed Glide virtual screening workflow (Glide VSW) that enables the docking of ligands against one or more targets simultaneously. Both Glide docking and Glide VSW include ligand preparation, the filtering of results and the ability to select three incremental stages of ranking accuracy: high-throughput virtual screening (Glide HTVS), standard precision (Glide SP) and extra precision level (Glide XP) [60]. HTVS precision level is a less stringent and the fastest docking method in Glide for efficiently enriching a million compound libraries. The main objective of this type of docking is to estimate the size and volume of ligands, and if the volume exceeds the volume of the binding site of the receptor, then the ligand is automatically excluded from further docking procedures [61]. However, such automatic exclusion of ligands from the sample pool can have a significant impact on the results. Glide SP performs a more forgiving evaluation of ligands that have a reasonable propensity to bind to a receptor, even if the predicted binding pose has significant imperfections [30]. The use of Glide SP as a separate module of Schrödinger Maestro software or as a second step in Glide VSW minimizes false negatives in the final set of compounds. However, even at this level of accuracy, Glide can automatically exclude some ligands from the sample pool due to their molecular size. Glide XP is usually the last stage of Glide VSW for the further elimination of false positives, accomplished by more extensive sampling and advanced scoring, resulting in higher enrichment of the final set of compounds. Glide XP performs semi-quantitative ranging of ligands by their ability to bind to a target protein, which is considered a rigid body. The main advantages of Glide XP are (I) the application of large desolvation penalties to both ligand and protein polar and charged groups in appropriate cases and (II) the identification of specific structural motifs that provide exceptionally large contributions to enhanced binding affinity [60].

## 2. Results and Discussions

### 2.1. Binding Energy and Ligand Efficiency as Selection Criteria

One of the first questions during the virtual screening results analysis is as follows: which criterion is better for the effective and successful selection of potential drug candidates? How effective is the binding energy or LE as a selection criterion?

In order to examine this, the dataset for each target protein was first docked to the respective active site using three selected molecular docking programs (AutoDock 4.2.6 [28], Vina 1.1.2 [29] and Glide [30,31,32]). Detailed molecular docking results for each molecular target are given in the Appendix A, i.e., AChE docking results—Appendix A (Appendix A); BACE1 docking results—Appendix A; GSK3β docking results—Appendix A; SERT central and allosteric binding sites—Appendix A and Appendix A, respectively; and TrkA docking results—Appendix A. After that, the molecular docking results for all target proteins were sorted and analyzed according to the binding energy and LE. Values of LE higher than 0.3 and values of the binding energy lower than −8.0 kcal/mol were used as currently accepted reference values for potentially active compounds in drug design [13,14,15].

For all studied docking programs, the composition of the final set of compounds, selected by binding energy (ΔG ≤ −8.0 kcal/mol), is almost identical for all target proteins, and the proportion of each group of compounds is approximately 30% (Figure 2). However, it is noticeable that the composition of the final set of compounds, as well as its size, depends not only on the type of target protein but also on the software used. Thus, it can be suggested that using the same binding energy threshold for different types of target proteins, as well as for different software, limits the efficient and successful selection of potential drug candidates. As can be seen in Figure 2, for TrkA, almost all compounds of the initial benchmark set of compounds can be selected as potentially high-active compounds based on their binding energy calculated by Vina 1.1.2 (148 compounds of 150 with ΔG ≤ −8.0 kcal/mol). In contrast to Vina 1.1.2, the use of such a threshold value of binding energy in the case of Glide SP or Glide XP docking for TrkA is more reliable and allows one to obtain a final set of compounds with a high number of “lead” compounds (pIC_50_ ≥ 8), namely 22 compounds of 41 (53%) for Glide SP and 22 compounds of 37 (59%) for Glide XP (Figure 2). Thus, it can be assumed that the binding energy of −8.0 kcal/mol is not a “universal” threshold value and should be selected according to the software used as well as the type of target protein. For instance, for TrkA, in the case of Vina 1.1.2, the use of binding energy lower than −10.0 kcal/mol as a selection criterion can help to significantly reduce the size of the final set of compounds and increase the number of “lead” compounds in it (49 compounds with ΔG ≤ −10.0 kcal/mol, of which 32 compounds (65%) with pIC_50_ ≥ 8) (Appendix A). It should be noted that, in the case of BACE1, the use of this threshold value of the binding energy (ΔG ≤ −8.0 kcal/mol) is a suitable selection criterion only for the Vina 1.1.2 docking results. The selection of compounds by their binding energy calculated by other studied programs (AutoDock 4.2.6 and Glide) failed (Figure 2). Based on these results, the binding energy threshold of −8.0 kcal/mol is not suitable for the reliable identification of potentially active compounds for AChE, GSK3β and SERT, and a much lower binding energy value should be used. Moreover, for target proteins such as BACE1 and TrkA, a higher binding energy should be used as a reference value for the reliable selection of potential drug candidates. For instance, in the case of AutoDock 4.2.6, an increase in the binding energy threshold value of up to −7.0 kcal/mol allows for the selection of a much larger number of potential hit compounds for BACE1 and TrkA, with 43 and 72 compounds, respectively, and at least 40% of this set are compounds with pIC_50_ ≥ 8 (Appendix A).

Further analysis of the docking results showed that the selection of compounds by LEs is very similar to the selection of compounds by binding energy (Figure 3). The ratio of the number of “lead” (pIC_50_ ≥ 8), “hit” (pIC_50_ = 6–5) and low-active (inactive, pIC_50_ ≤ 4) compounds in the final set of compounds remains at the level of 1:3. Nevertheless, for some targets, the use of LE may be more efficient than using binding energy as a selection criterion. For example, for GSK3β, the selection of compounds by LE allowed for a reduction in the number of low-active (inactive) compounds in the final set of compounds by almost two times (Figure 2 and Figure 3).

It was also seen that, for TrkA, in the case of Glide SP and Glide XP, the number of highly active compounds in the final set increased significantly, from 53% to 72% of “lead” compounds in final set. Thus, in some cases, the LE can be considered a more efficient selection criterion. However, regardless of which parameter is used to select potential drug candidates (the binding energy or LE), the threshold value should be selected individually for each task.

### 2.2. Identification of a “Reliable Range” of Binding Energy and LE

Another important question that researchers face in computational modeling is if there is a “reliable range” for the docking binding free energy or LE. Additionally, based on this, would it be possible to select compounds with biological activity in the desired concentration range?

Further analysis was focused on the evaluation of docking results by the best and poorest values of binding energy and LE, as well as their average values for each group of compounds and for each molecular target separately.

#### 2.2.1. AutoDock 4.2.6

According to the AutoDock 4.2.6 docking results, for each target, the calculated binding energy values are within the same range for each group of compounds (Table 2), and consequently, it is not possible to reliably filter highly active compounds from low-active or inactive compounds. However, further evaluation of the average binding energy (Table 2) showed that, for some of the studied targets, it is possible to suggest the most optimal binding energy at which a compound is more likely to be biologically active. Thus, the selection of compounds with binding energies below −11.0 kcal/mol for AChE, below −9.0 kcal/mol for GSK3β and below −7.0 kcal/mol for TrkA can increase the proportion of compounds with pIC_50_ higher than 8 in the final set of compounds (Table 2). It should be noted that for AChE, the standard deviation of the binding energy for the 1st and 3rd groups of compounds is relatively high compared to those for other targets. Therefore, it can be assumed that the identified binding energy threshold value for AChE does not guarantee that all selected compounds are highly active, but it can be a useful criterion for the initial filtration of HTVS results. Unfortunately, based on the data obtained, it is difficult to establish a similar threshold average value of the binding energy for highly active compounds for BACE1, GSK3β and SERT (both active sites). It can be speculated that, in the case of BACE1, compounds with binding energies higher than −5.0 kcal/mol are more likely to be inactive or to have their pIC_50_ lower than 4. In addition, it can be assumed that biological activity in the range of pIC_50_ up to 5 can be expected for compounds with binding energies lower than −6.0 kcal/mol. For GSK3β, a binding energy below −8.0 kcal/mol can be suggested as a threshold value for the filtration of the initial library. In the case of the central binding site of SERT, there is no correlation between binding energy and biological activity. In the case of the allosteric binding site of SERT, it can be said that the calculated binding energy is inversely proportional to the biological activity of the compounds. Therefore, it can be assumed that docking to the central binding site can likely lead to more informative results despite the lack of correlation between binding energy and biological activity (Table 2).

Further evaluation of the docking results showed that, for all targets, the obtained best and poorest LE values are almost identical for each group of compounds. They do not allow for ranking compounds by their biological activity and, accordingly, cannot provide a reliable ranking of compounds by their biological activity (Table 2). This assumption can also be supported by the fact that the best LE value for each target was obtained for a low-active compound, namely compound CHEMBL77510 with a pIC_50_ of 3.252 for the AChE dataset (Appendix A), compounds CHEMBL3261081 and CHEMBL3261080 with pIC_50_ values of 3.102 and 2.086 for the BACE1 dataset (Appendix A), respectively, compound CHEMBL608419 with a pIC_50_ of 4.000 for the GSK3β dataset (Appendix A), compound CHEMBL1173276 with a pIC_50_ of 4.000 for the SERT dataset (Appendix A and Appendix A) and compound CHEMBL3629609 with a pIC_50_ of 3.851 for the TrkA dataset (Appendix A). The calculated average values of LE also support this observation (Table 2).

#### 2.2.2. AutoDock Vina 1.1.2

According to the results obtained for all targets using Vina 1.1.2, the calculated intervals of the binding energy are almost identical for each group of compounds (Table 3). Therefore, as in the case of AutoDock 4.2.6, the calculated binding energy can be used only for the initial filtration of HTVS results. Only in the case of AChE and BACE1, a trend towards better (more negative value) average calculated binding free energies can be observed.

Similar to the AutoDock 4.2.6 results, the LE values calculated using Vina 1.1.2 predicted binding energies also cannot provide a reliable ranking of compounds according to their pIC_50_ range (Table 3). However, for all protein targets except the SERT allosteric site, the best (highest) LE values were obtained for compounds with pIC_50_ values lower than 4 (Appendix A).

#### 2.2.3. Glide HTVS

As mentioned above, the main task of Glide HTVS docking is the fast evaluation and exclusion of ligands whose sizes and volumes exceed the volume of the binding site of the receptor [61]. Thus, it can be assumed that the results of Glide HTVS may strongly depend on the size and structure of the active site of the receptor. In our case, the largest number of automatically excluded compounds was obtained for proteins with very narrow and hard-to-reach active sites: AChE and the central binding site of SERT, respectively. In the case of the central binding site of SERT, 41 compounds (27% of ligands) were excluded from the docking procedure, and the number for the allosteric site was significantly lower (Appendix A). Such an observation may suggest that, in the development of new SERT inhibitors, both sites should be considered. Otherwise, the primary filtering of compounds using Glide HTVS can lead to the loss of some biologically active compounds. For AChE, as a protein with a very narrow and specific active site, one would expect more compounds to be excluded, but only 12% of the compounds were excluded during Glide HTVS docking. This result can be explained by the fact that most ligands likely bind during docking to the peripheral anionic binding subsite (PAS) of AChE, which is located close to the protein surface and provides further binding of the ligand to the active site [62]. Likely, the PAS plays the role of the “main” binding site during the docking procedure and, thus, compensates for the inaccessibility of the catalytic active site. The large difference in the number of excluded compounds for the central and allosteric sites of SERT confirms this assumption. For the SERT allosteric site that is located close to the surface and that is more accessible for ligand binding, the number of excluded compounds is more than ten times less than that in the case of the central binding site. For GSK3β, a protein with a relatively wide active center close to the protein surface, only two compounds were automatically excluded from the docking procedure.

Further evaluation of calculated Glide HTVS binding energies and LEs demonstrated that, for all targets, the average values of both parameters are almost identical for each group of compounds studied (Table 4). Keeping in mind that Glide HTVS is a program for the rapid evaluation of ligand binding, this result is expected. This observation once again confirms that the HTVS precision level can only be used for the reduction in the initial library size. The observation that one compound from the AChE benchmark set gave a positive binding energy value, even when it has experimentally significant inhibitory activity against AChE, further supports this assumption (CHEMBL4168179 with a pIC_50_ of 5.087, Appendix A).

#### 2.2.4. Glide SP

Despite a more lenient compound binding evaluation algorithm than Glide HTVS, Glide SP can also automatically exclude compounds from the sample pool. For AChE, the number of automatically excluded compounds compared with the docking results obtained by Glide HTVS remained high (18 and 9 compounds for HTVS and SP, respectively) (Appendix A), whereas for other studied target proteins, this number is significantly lower. Obviously, the number of excluded compounds directly depends on the specificity and complexity of the active site of the target protein. The intervals of calculated Glide SP binding energies for each studied target protein and for each group of compounds are very close and do not suggest of any “reliable range” for the selection of compounds with activity in the desired concentration range (Table 5). Further evaluation of the average binding energies was more informative and allowed us to identify the threshold binding energy for one of the studied targets that can be used for the selection of potential “lead” and “hit” compounds. Thus, for AChE, the use of a binding energy lower than −8.5 kcal/mol as a reference value can likely help reduce the number of low-active or inactive compounds in the final dataset. The average binding energy for TrkA suggests that compounds with binding energies lower than −8.0 kcal/mol are likely to be highly active (pIC_50_ ≥ 8) in biological experiments. The average binding energies obtained for BACE1, SERT (both binding site) and GSK3β are very similar for each group of compounds and can be used only for the preliminary filtration of docking results (Table 5).

The further analysis of LE values also did not identify any reliable LE value for the selection of compounds with a given biological activity (Table 5).

#### 2.2.5. Glide XP

As in the case of Glide HTVS and SP, during the processing of obtained Glide XP results, we came across the fact that some compounds from the benchmark set were automatically excluded from the sample pool. Usually, the reasons for this exclusion at this precision level are as follows: (I) a poor fit for the ligand in the active site, and high-energy clashes cannot be resolved in the minimization step, and (II) the ligand can be rejected because of VdW clashes. The rejection of ligands due to VdW clashes can be avoided by increasing the Coulomb–VdW cutoff to a large, positive value. However, in our experience, increasing the Coulomb–VdW cutoff up to 350 kcal/mol during the docking of another benchmark set of active compounds to the active site of AChE did not completely avoid the automatic exclusion of compounds from the sample pool, but it helped to significantly reduce their number compared with those for Glide HTVS and SP (data are not shown). For other studied target proteins, the number of excluded compounds was also significantly lower than those in the case of Glide HTVS and SP, but this improvement in results was only due to the improved scoring function of Glide XP. The problem of the automatic exclusion can also be resolved by Induced Fit Docking [63]; however, this approach is rarely used in the early stages of new drug development.

The calculated Glide XP binding energies also provide sufficient information for the enrichment of the compound library but do not provide enough information for the reliable ranking of compounds by their biological activity (Table 6). As in the case of Glide HTVS and SP, the binding energies calculated using the Glide XP vary within a wide range. Moreover, in the case of BACE1, positive binding energies were obtained for some compounds (Appendix A).

The calculated average values of LE are also almost identical for each group of compounds (Table 6). The obtained LE intervals for each target and each group of compounds start from extremely low values, and the best LE values were obtained for compounds with moderate or low biological activity (Appendix A). The average values of LE also remain noticeably lower than the corresponding values obtained with AutoDock 4.2.6 and Vina 1.1.2. Accordingly, when processing docking results obtained with Glide XP, using LE as a selection criterion may cause a large number of really active compounds to be excluded.

## 3. Materials and Methods

### 3.1. Target Proteins and Compound Library

#### 3.1.1. Protein Structure Preparation

In this work, the X-ray structures of the recombinant human AChE (PDB ID: 4EY6) [36], human BACE1 (PDB ID: 6EQM) [39], human GSK3β (PDB ID: 1PYX) [41], ts3 human SERT (PDB IDs: 5I6X and 5I73) [43] and human TrkA (PDB ID: 4AOJ) [45] were downloaded from Protein Data Bank [64]. Before the molecular docking procedure, the raw crystal structures of all studied target proteins were treated using Schrödinger Protein Preparation Wizard [65,66].

#### 3.1.2. Compound Libraries

The data on the biologically active compounds against the studied target proteins were extracted from the ChEMBL database [67]. The obtained datasets were prepared as follows: (I) newer experimental data were preferable; (II) where possible, the same or similar experimental protocol for activity determination was used; and (III) compounds with diverse structures were selected. As a result, the benchmark set containing 150 compounds was constructed for each target protein. Thereafter, each benchmark set was divided into 3 groups (50 compounds each) according to their pIC_50_ [M]:-1st group “lead” compounds—pIC_50_ ≥ 8 (IC_50_ ≤ 10 nM);-2nd group “hit” compounds—pIC_50_ = 6–5 (IC_50_ = 1–10 μM);-3rd group low-active compounds with pIC_50_ ≤ 4 (IC_50_ ≥ 100 μM) or inactive compounds.

The two-dimensional chemical structures of small-molecule ligands were prepared and optimized before docking using the Schrödinger Maestro LigPrep procedure [68]. The OPLS4 force field [69] was used in all ligand preparation steps. PDB files were created from the lowest energy conformers for each ligand. The PDBQT files were generated using Raccoon v.1.0f [55].

### 3.2. Molecular Docking

#### 3.2.1. Receptor Grid Generation

The binding interfaces between the co-crystallized ligand and receptor for each target protein were identified using the Schrödinger Glide Grid Generation procedure [30,31]. The identified coordinates of active sites were each used for docking with the molecular docking software.

#### 3.2.2. AutoDock 4.2.6

AutoDock 4.2.6 [28] was used for the docking of the benchmark set of compounds to the active sites of the studied target proteins. The molecular docking procedure was performed in batch mode using Raccoon v.1.0f [55] generated and in-house optimized scripts. Molecular docking was carried out using the default settings. Those were defined as follows: 100 GA runs, a population size of 150, a maximum number of evaluations of 2.5 × 10^6^, a maximum number of generations of 2.7 × 10^4^, a maximum number of top individuals that automatically survive of 1 and a rate of gene mutations of 0.02.

#### 3.2.3. AutoDock Vina 1.1.2

The binding energy of the selected compounds to the studied target proteins was also determined using AutoDock Vina 1.1.2 [29]. The docking parameters were used in their default values, as follows: 1CPU to use, an exhaustiveness of 8 and 9 poses to output. AutoDock Vina 1.1.2 docking was carried out automatically in batch mode using scripts written in-house.

#### 3.2.4. Glide Docking

The selected compounds were also docked to the active site of each of the studied target proteins using the Schrödinger’s Glide docking procedure [30,31]. Docking was carried out using each of three stages of ranking accuracy (HTVS, SP and XP). The docking parameters were used at their default values.

#### 3.2.5. Ligand Efficiency Calculation

In this work, for all compounds, the LE was calculated as follows:(1)$$LE=\frac{\Delta G_{bind}}{N_{h}}$$
where *N_h_* denotes the number of non-hydrogen atoms in the small-molecule ligand.

## 4. Conclusions

In this work, we analyzed the impact of the docking software as well as the type of the target protein on the molecular docking results and its accuracy on an example of the three most popular docking programs and five target proteins related to neurodegenerative diseases. In addition, we attempted to determine a “reliable range” for binding energy and LE which would allow for the selection of compounds with biological activity in the desired concentration range. The obtained data suggest that the use of the same binding energy or LE threshold for different types of target proteins, as well as for different software, limits the efficient and successful selection of potential drug candidates. Consequently, different threshold values are necessary for each target protein when using the binding energy or LE as a selection criterion for potentially active compounds. It was also shown that results and threshold values of the binding energy or LE are highly dependent on the selection of the docking software and the type of target protein. According to our results, it can be assumed that, in the case of target proteins with narrow and hard-to-reach active sites, such as in the case of AChE, the selection of the search algorithm and scoring function can have a great impact on the virtual screening results and, in some cases, can cause the exclusion of a large number of potentially active compounds. In addition, in the SERT example, it was demonstrated that it is necessary to take into account all active sites present in the protein. Thus, the rational selection of docking software and selection criteria, as well as the consideration of the features and characteristics of the target protein and the use of several parameters simultaneously for the selection of potentially active compounds, can help improve the efficiency of the early stage of drug development.

## Figures and Tables

**Figure 1 molecules-27-09041-f001:**
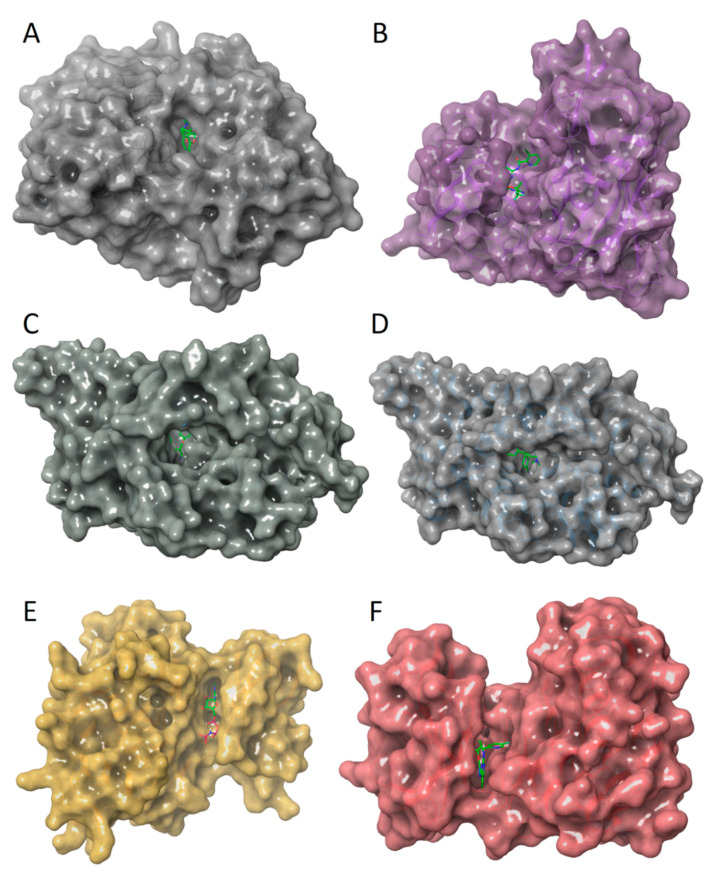
Selected target proteins: (**A**) human AChE in complex with (-)-galantamine (PDB ID: 4EY6); (**B**) human BACE1 in complex with inhibitor CNP520 (PDB ID: 6EQM); (**C**) ts3 human SERT in complex with paroxetine at the central site (PDB ID: 5I6X); (**D**) ts3 human SERT in complex with *S*-citalopram at the allosteric site (PDB ID: 5I73); (**E**) human GSK3β in complex with non-selective ATP-mimetic inhibitor (PDB ID: 1PYX); (**F**) human TrkA in complex with the inhibitor AZ-23 (PDB ID: 4AOJ). All target proteins are shown as a molecular surface, and the co-crystallized ligands are stick models colored green (carbon), blue (nitrogen), red (oxygen) and purple (phosphorus).

**Figure 2 molecules-27-09041-f002:**
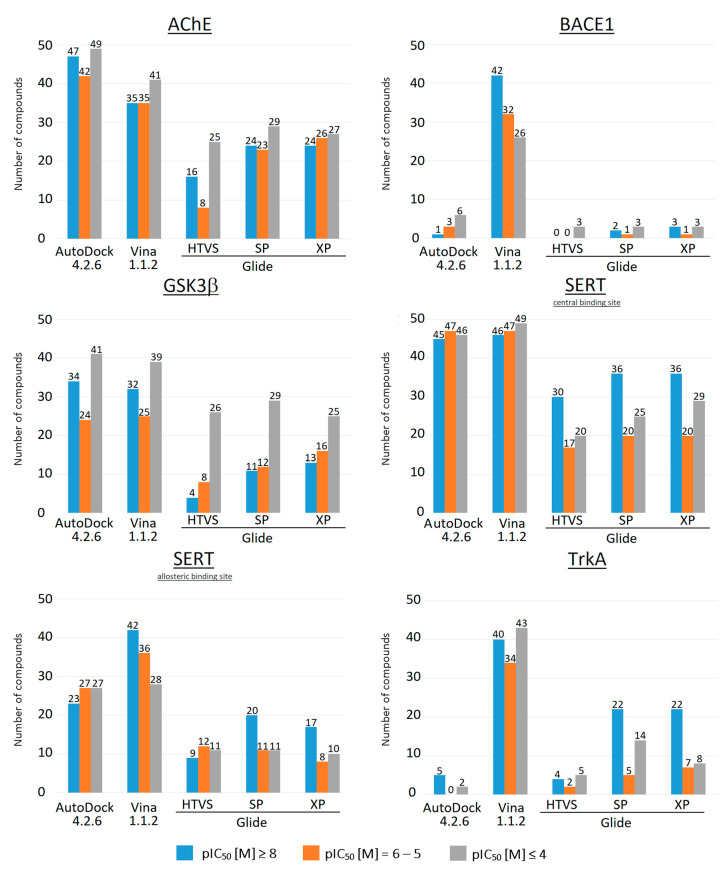
Number of compounds selected by calculated binding energy (ΔG ≤ 8.0 kcal/mol) for all studied target proteins.

**Figure 3 molecules-27-09041-f003:**
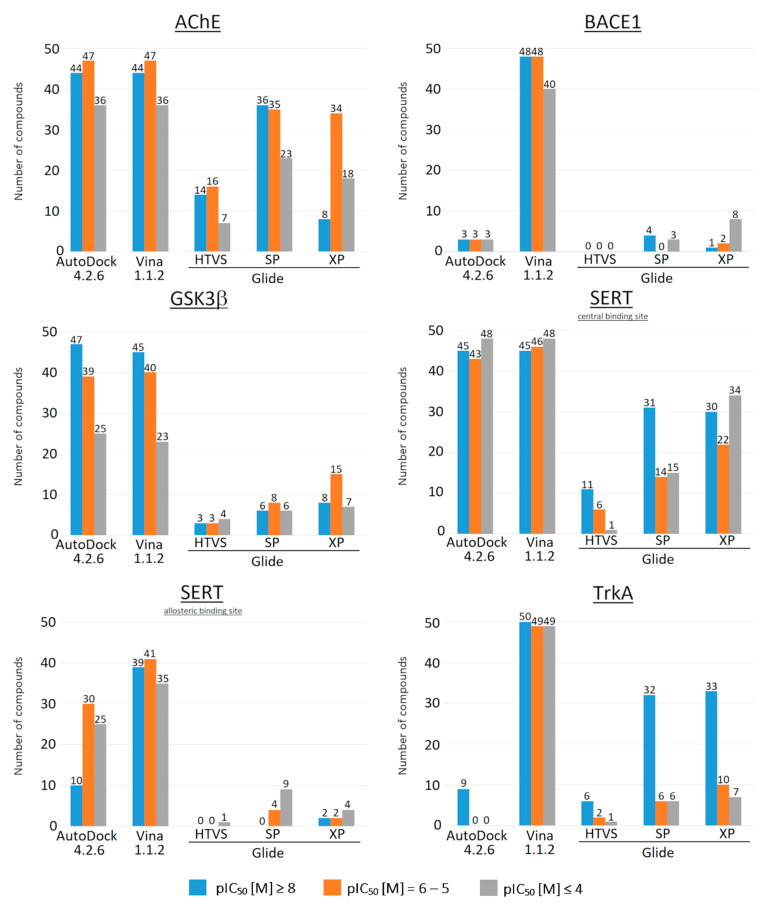
Number of compounds selected by ligand efficiency (LE ≥ 0.3) for all studied target proteins.

**Table 1 molecules-27-09041-t001:** The list of the studied target proteins.

Target	Classification	Disease	Ref.	PDB ID	Ref.
AChE	Hydrolase	AD	[34,35]	4EY6	[36]
BACE1	Hydrolase		[37,38]	6EQM	[39]
GSK3β	Transferase		[40]	1PYX	[41]
SERT	Transport protein		[42]	5I6X	[43]
				5I73	
TrkA	Transferase	PD	[44]	4AOJ	[45]

**Table 2 molecules-27-09041-t002:** The best (lowest binding energy), poorest (highest binding energy) and average values of calculated AutoDock 4.2.6 binding energies (ΔG, kcal/mol) and ligand efficiencies (LE). The average values of binding energies and LEs are presented as mean ± SD.

Target	pIC_50_	ΔG, kcal/mol			LE		
		Best	Poorest	Average	Best	Poorest	Average
AChE	≥8	−16.16	−4.91	−11.88 ± 2.70	0.61	0.23	0.39 ± 0.07
	6–5	−14.64	−7.58	−10.89 ± 1.76	0.46	0.21	0.36 ± 0.06
	≤4	−15.59	−5.81	−9.94 ± 2.55	0.67	0.26	0.40 ± 0.06
BACE1	≥8	−8.88	−3.29	−6.65 ± 1.00	0.31	0.07	0.22 ± 0.04
	6–5	−8.67	−4.50	−6.38 ± 1.09	0.40	0.08	0.21 ± 0.06
	≤4	−8.60	−0.96	−5.25 ± 1.93	0.46	0.02	0.24 ± 0.12
GSK3β	≥8	−11.84	−6.74	−9.47 ± 1.14	0.51	0.20	0.39 ± 0.06
	6–5	−12.29	−6.94	−10.10 ± 1.31	0.53	0.28	0.38 ± 0.06
	≤4	−12.40	−7.34	−9.99 ± 1.38	0.56	0.28	0.38 ± 0.08
SERT Central binding site	≥8	−11.15	−5.05	−7.01 ± 1.31	0.39	0.16	0.29 ± 0.05
	6–5	−10.89	−5.92	−8.35 ± 1.19	0.41	0.22	0.31 ± 0.05
	≤4	−10.72	−6.00	−8.38 ± 1.50	0.42	0.25	0.31 ± 0.04
SERT Allosteric binding site	≥8	−12.15	−6.80	−9.25 ± 1.02	0.42	0.23	0.32 ± 0.05
	6–5	−11.62	−6.75	−8.89 ± 1.25	0.48	0.22	0.31 ± 0.06
	≤4	−11.71	−5.19	−8.34 ± 1.33	0.52	0.21	0.35 ± 0.07
TrkA	≥8	−8.90	−5.89	−7.50 ± 0.62	0.32	0.18	0.24 ± 0.03
	6–5	−7.89	−4.85	−6.63 ± 0.75	0.29	0.10	0.22 ± 0.05
	≤4	−7.92	−4.28	−6.76 ± 0.74	0.38	0.12	0.24 ± 0.04

**Table 3 molecules-27-09041-t003:** The best (lowest binding energy), poorest (highest binding energy) and average values of calculated AutoDock Vina 1.1.2 binding energies (ΔG, kcal/mol) and ligand efficiencies (LE). The average values of binding energies and LEs are presented as mean ± SD.

Target	pIC_50_	ΔG, kcal/mol			LE		
		Best	Poorest	Average	Best	Poorest	Average
AChE	≥8	−12.70	−5.90	−10.46 ± 1.64	0.74	0.18	0.35 ± 0.10
	6–5	−11.90	−7.50	−9.83 ± 1.16	0.45	0.19	0.33 ± 0.06
	≤4	−11.30	−5.70	−8.84 ± 1.51	0.69	0.24	0.37 ± 0.09
BACE1	≥8	−11.90	−7.50	−10.06 ± 1.08	0.43	0.16	0.33 ± 0.05
	6–5	−11.50	−6.10	−9.61 ± 1.04	0.47	0.19	0.32 ± 0.07
	≤4	−11.50	−4.80	−8.63 ± 1.11	0.68	0.15	0.37 ± 0.15
GSK3β	≥8	−12.70	−7.50	−9.60 ± 1.16	0.51	0.25	0.39 ± 0.06
	6–5	−12.10	−5.90	−9.86 ± 1.39	0.52	0.14	0.37 ± 0.07
	≤4	−12.80	−6.80	−10.09 ± 1.61	0.51	0.28	0.38 ± 0.06
SERT Central binding site	≥8	−10.80	−6.60	−8.72 ± 0.99	0.50	0.21	0.36 ± 0.06
	6–5	−10.40	−6.50	−8.88 ± 0.98	0.49	0.23	0.33 ± 0.06
	≤4	−10.80	−6.30	−8.89 ± 1.22	0.44	0.25	0.33 ± 0.06
SERT Allosteric binding site	≥8	−10.90	−6.90	−8.90 ± 0.69	0.45	0.24	0.31 ± 0.05
	6–5	−10.90	−6.60	−8.86 ± 1.13	0.47	0.22	0.31 ± 0.06
	≤4	−10.20	−5.10	−8.18 ± 1.06	0.55	0.21	0.35 ± 0.08
TrkA	≥8	−11.60	−8.70	−10.08 ± 0.59	0.42	0.23	0.33 ± 0.04
	6–5	−10.70	−7.80	−9.48 ± 0.63	0.42	0.19	0.31 ± 0.05
	≤4	−11.90	−8.20	−9.34 ± 0.73	0.51	0.23	0.34 ± 0.06

**Table 4 molecules-27-09041-t004:** The best (lowest binding energy), poorest (highest binding energy) and average values of calculated Glide HTVS binding energies (ΔG, kcal/mol) and ligand efficiencies (LE). The average values of binding energies and LEs are presented as mean ± SD.

Target	pIC_50_	ΔG, kcal/mol			LE		
		Best	Poorest	Average	Best	Poorest	Average
AChE	≥8	−11.11	−2.41	−7.30 ± 1.61	0.64	0.05	0.27 ± 0.11
	6–5	−10.76	1.74	−7.14 ± 2.15	0.36	0.07	0.25 ± 0.07
	≤4	−9.95	−2.02	−6.65 ± 1.58	0.68	0.06	0.30 ± 0.12
BACE1	≥8	−7.79	−1.56	−4.27 ± 1.33	0.27	0.05	0.14 ± 0.05
	6–5	−6.67	−0.34	−3.84 ± 1.37	0.26	0.01	0.14 ± 0.06
	≤4	−6.57	−1.10	−4.34 ± 1.09	0.55	0.02	0.20 ± 0.11
GSK3β	≥8	−9.67	−4.24	−7.41 ± 1.05	0.44	0.18	0.33 ± 0.07
	6–5	−9.50	−4.46	−6.86 ± 1.17	0.44	0.13	0.29 ± 0.09
	≤4	−8.41	−4.26	−7.04 ± 0.85	0.46	0.16	0.33 ± 0.08
SERT Central binding site	≥8	−7.63	−4.06	−5.92 ± 0.85	0.36	0.10	0.25 ± 0.06
	6–5	−7.79	0.19	−5.89 ± 1.42	0.37	0.01	0.23 ± 0.08
	≤4	−9.40	−4.98	−6.28 ± 0.80	0.35	0.13	0.24 ± 0.06
SERT Allosteric binding site	≥8	−10.06	−3.60	−6.02 ± 1.09	0.37	0.09	0.22 ± 0.06
	6–5	−8.50	−3.12	−6.01 ± 1.36	0.46	0.07	0.22 ± 0.09
	≤4	−8.79	−3.01	−5.95 ± 1.36	0.56	0.10	0.27 ± 0.10
TrkA	≥8	−10.03	−3.51	−6.09 ± 1.58	0.35	0.10	0.20 ± 0.06
	6–5	−8.48	−3.71	−5.91 ± 1.29	0.33	0.11	0.20 ± 0.05
	≤4	−8.04	−2.78	−5.94 ± 1.17	0.38	0.09	0.22 ± 0.06

**Table 5 molecules-27-09041-t005:** The best (lowest binding energy), poorest (highest binding energy) and average values of calculated Glide SP binding energies (ΔG, kcal/mol) and ligand efficiencies (LE). The average values of binding energies and LEs are presented as mean ± SD.

Target	pIC_50_	ΔG, kcal/mol			LE		
		Best	Poorest	Average	Best	Poorest	Average
AChE	≥8	−11.38	−5.62	−8.96 ± 1.50	0.70	0.14	0.31 ± 0.09
	6–5	−10.71	−5.78	−8.72 ± 1.08	0.41	0.19	0.29 ± 0.05
	≤4	−10.42	−4.62	−7.79 ± 1.38	0.61	0.18	0.33 ± 0.10
BACE1	≥8	−10.16	−3.99	−5.85 ± 1.53	0.38	0.11	0.19 ± 0.06
	6–5	−7.73	−2.91	−5.07 ± 1.29	0.30	0.08	0.17 ± 0.06
	≤4	−8.95	−3.32	−5.55 ± 1.33	0.62	0.10	0.24 ± 0.11
GSK3β	≥8	−9.94	−4.85	−8.16 ± 1.07	0.47	0.12	0.35 ± 0.07
	6–5	−9.58	−1.92	−7.54 ±1.11	0.51	0.05	0.29 ± 0.09
	≤4	−9.07	−4.54	−7.59 ± 0.88	0.46	0.13	0.30 ± 0.09
SERT Central binding site	≥8	−7.75	−4.81	−6.59 ± 0.69	0.39	0.13	0.27 ± 0.06
	6–5	−10.09	−4.89	−6.68 ± 0.90	0.38	0.16	0.25 ± 0.06
	≤4	−9.84	−5.70	−6.98 ± 1.17	0.38	0.15	0.26 ± 0.06
SERT Allosteric binding site	≥8	−13.00	−5.45	−7.12 ± 1.22	0.41	0.16	0.25 ± 0.06
	6–5	−11.65	−4.80	−6.99 ± 1.38	0.65	0.11	0.26 ± 0.10
	≤4	−11.97	−4.44	−6.86 ± 1.64	0.54	0.16	0.30 ± 0.10
TrkA	≥8	−10.67	−3.83	−8.61 ± 1.76	0.40	0.13	0.28 ± 0.07
	6–5	−9.62	−4.02	−6.72 ± 1.19	0.40	0.08	0.22 ± 0.06
	≤4	−9.38	−2.16	−6.73 ± 1.33	0.44	0.08	0.25 ± 0.07

**Table 6 molecules-27-09041-t006:** The best (lowest binding energy), poorest (highest binding energy) and average values of calculated Glide XP binding energies (ΔG, kcal/mol) and ligand efficiencies (LE). The average values of binding energies and LEs are presented as mean ± SD.

Target	pIC_50_	ΔG, kcal/mol			LE		
		Best	Poorest	Average	Best	Poorest	Average
AChE	≥8	−11.77	−4.68	−9.44 ± 2.06	0.02	0.70	0.31 ± 0.12
	6–5	−13.10	−2.82	−8.79 ± 2.07	0.09	0.44	0.29 ± 0.08
	≤4	−15.80	−4.23	−7.54 ± 2.04	0.11	0.69	0.32 ± 0.12
BACE1	≥8	−11.61	−2.65	−5.08 ± 1.75	0.07	0.43	0.16 ± 0.06
	6−5	−8.11	4.20	−4.09 ± 2.39	0.02	0.36	0.15 ± 0.07
	≤4	−9.85	−2.49	−5.46 ± 2.03	0.07	0.51	0.22 ± 0.08
GSK3β	≥8	−11.34	−0.73	−8.19 ± 1.57	0.02	0.46	0.34 ± 0.09
	6–5	−10.72	−4.30	−7.85 ± 1.24	0.18	0.45	0.30 ± 0.08
	≤4	−10.00	−6.33	−8.40 ± 0.83	0.17	0.48	0.32 ± 0.09
SERT Central binding site	≥8	−8.60	−2.54	−6.24 ± 1.20	0.06	0.35	0.26 ± 0.06
	6–5	−8.67	−1.84	−5.68 ± 1.48	0.05	0.35	0.22 ± 0.07
	≤4	−9.82	−2.83	−5.93 ± 1.41	0.07	0.38	0.23 ± 0.08
SERT Allosteric binding site	≥8	−13.98	−3.80	−6.67 ± 1.80	0.10	0.41	0.24 ± 0.08
	6–5	−11.63	−3.15	−6.93 ± 1.89	0.10	0.58	0.25 ± 0.11
	≤4	−14.38	−2.73	−6.81 ± 2.36	0.08	0.64	0.30 ± 0.13
TrkA	≥8	−11.41	−2.64	−8.46 ± 2.35	0.09	0.42	0.28 ± 0.09
	6–5	−10.35	−1.88	−6.28 ± 2.03	0.04	0.38	0.21 ± 0.08
	≤4	−9.39	−3.04	−6.26 ± 1.45	0.11	0.40	0.23 ± 0.07

## Data Availability

Not applicable.

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
