# Peer review of "The Impact of Software Used and the Type of Target Protein on Molecular Docking Accuracy"

_molecules, 2022, doi:10.3390/molecules27249041_

Round 1
Reviewer 1 Report
Recommendation: The manuscript entitled “The Impact of Software Used and Type of Target Protein on Molecular Docking Accuracy” could be publishable and further review is needed.
In this manuscript, authors reported that the essence of various molecular docking tools (AutoDock 4.2.6, AutoDock Vina 1.1.2, and GLIDE) with five different type of the target proteins related to neurodegenerative diseases. This paper is well written and they also selected numerous biologically active compounds from ChEMBL databases. The selection of targets and compounds are sufficient to study the importance of molecular docking tool for the drug design and development purposes. Overall, I recommend publication, but I have some concerns that the authors should address:
· Authors should mention the ligand efficiency formula in the revised manuscript.
· Authors should explain the merits and demerits of OPLS and other force field and this should be validate by other commercial software’s
· Author should justify the how can docking only decides the efficiency of the target?
· Authors should confirm the free energy profile obtained from MD simulations and the trend in the binding score and free energy should be compared with the available list of complexes.
· Grammatical check is required for the manuscript.
Author Response
First, we would like to thank the reviewer for their comments and suggestions. The detailed answers to the reviewer are listed below:
- Authors should mention the ligand efficiency formula in the revised manuscript. Answer: The information about ligand efficiency calculations is currently added to the materials and method section (p. 16, lines 533-536).
- Authors should explain the merits and demerits of OPLS and other force field and this should be validate by other commercial software’s. Answer: The comparison of the advantages and disadvantages of the different types of force fields was not the aim of the present study.
- Author should justify the how can docking only decides the efficiency of the target? Answer: In our manuscript, we do not claim that molecular docking can provide sufficient information about the efficiency of a compound or target.
- Authors should confirm the free energy profile obtained from MD simulations and the trend in the binding score and free energy should be compared with the available list of complexes. Answer: In the context of this work, molecular dynamics (MD) simulations were not carried out. The study of the correlation between energy calculated by docking and free energy calculated based on the MD simulation is the topic of a separate study.
- Grammatical check is required for the manuscript. Answer: We have gone once more through the English checking. It should be also mentioned that we are going to use the MDPI English checking service in order to improve the quality of the manuscript.
Reviewer 2 Report
1. In the abstract and paper the authors mentioned " docking results and its accuracy using the example of three most popular programs and five target proteins related to neurodegenerative diseases" Did you use these programs with any synthetic organic molecules and those get good results? related to any biological assay?
2. Recently many organic molecules have been reported with some docking studies related to mentioned programs. Why don't you mention this in the introduction part with an appropriate explanation? for example, you can use the following references with explanations like docking studies related to organic molecules.... how support docking program to this work like
1) Molecular Diversity, 2022, 26 (5), 2893-2905
2) 2020, In: Bioorganic Chemistry. 100, 9 p., 103908
3) 2022, In: RSC Advances. 12, 35, p. 22476-22491
4) Veligeti, R., Madhu, R. B., Anireddy, J., Pasupuleti, V. R., Avula, V. K. R., Ethiraj, K. S., Uppalanchi, S., Kasturi, S., et al., 2020, In: Scientific Reports. 10, 1, 22 p., 20720
Author Response
- In the abstract and paper the authors mentioned " docking results and its accuracy using the example of three most popular programs and five target proteins related to neurodegenerative diseases" Did you use these programs with any synthetic organic molecules and those get good results? related to any biological assay? Answer: We thank the referee for this question. We would like to draw the attention of the reviewer to the fact that all compounds of the benchmark dataset are “synthetic organic molecules”. All these compounds (small-molecule ligands) were obtained from the ChEMBL database. This information is indicated on p.15, lines 488-489, section 3.2.1. For all of them, the results of biological assays are reported. Additional information such as a biological activity measurement method and reference to the original publication can be found in the ChEMBL database using compound ChEMBL code that is indicated in the supplementary tables S1-S6.
- Recently many organic molecules have been reported with some docking studies related to mentioned programs. Why don't you mention this in the introduction part with an appropriate explanation? for example, you can use the following references with explanations like docking studies related to organic molecules.... how support docking program to this work like
1) Molecular Diversity, 2022, 26 (5), 2893-2905
2) 2020, In: Bioorganic Chemistry. 100, 9 p., 103908
3) 2022, In: RSC Advances. 12, 35, p. 22476-22491
4) Veligeti, R., Madhu, R. B., Anireddy, J., Pasupuleti, V. R., Avula, V. K. R., Ethiraj, K. S., Uppalanchi, S., Kasturi, S., et al., 2020, In: Scientific Reports. 10, 1, 22 p., 20720. Answer: We thank the reviewer for the recommendation of references. Molecular docking is a very widely used tool in modern drug design and a large number of articles are published on this topic every year. For example, a search for the keyword "molecular docking" in the Web of Science shows that 11,935 articles were published during the period from January to November (30th November) of this year. Unfortunately, referencing them all is an impossible task. Also, we got acquainted with the publications suggested by the reviewer and found that, unfortunately, none of the articles is in any way related to the subject of the current manuscript.
Reviewer 3 Report
General comment
The manuscript is interesting and tries to bridge the gap between the prediction of biologically active compounds using computational methods and the actual biological activity observed in the in vitro testing. The manuscript proposed that the gap can be minimized by setting up a threshold for which the prediction output becomes worthwhile for subsequent screening and/or analysis. This is a good idea. Below is some refinement to make it stronger for consideration upon its adoption. A scheme of the drug screening pipeline would be helpful to explain the state of the art and how this work contributes to improving it.
-
The application would be more appealing for use if there is an input for prior knowledge of the protein. For example, the protein X (the target) is known to have affinities towards K, L, M, and N types of compounds. An example of this is well provided in 2.1.2. This strategy should be made more profound.
-
It should be mentioned whether this would work to evaluate compounds that have been known or also for new compounds.
-
Considering that the main focus of the manuscript is screening and/or evaluation of potential molecules for drug development, it is surprising that Lipinski's rule is not mentioned at all because drug development would involve ADME at some points.
-
The molecular docking study in the manuscript employed several proteins. The protein structural model used are varying in resolution quality: AChE is 2.40Å, BACE1 is 1.35Å, GSK-3β is 2.40Å, TS3 is 3.14Å (5I6X) and 3.24Å (5I73), and TrkA 2.75Å. Protein structures themselves are models, thus there is an indigenous error (which can be checked through their R factors) and the resolution of the structure affects docking precision, especially when comparing the binding energies.
Detailed comments
line 25-90. Paragraph 1 is very long, it can be divided into several paragraphs.
Line 66-67. Please make clear what basic parameters were modified.
Line 69-71. It is confusing. There are 10 docking programs tested, and the same docking tool (does it mean one of the programs?) gives different outputs for the different protein families. Please clarify.
Line 76. Using quantum mechanics is said to be more accurate. However, usually, quantum mechanics is calculated in a vacuum field, which may not be appropriate to calculate the energy. Please explain.
Line 98-160. (2.1. Selection of biological targets and compounds library) and lines 161-247. (2.2. Molecular docking software). These parts are better distributed into the introduction and material and methods because it is hardly present and discuss the results of the experiment. Another option is: separate the result and discussion into two sections and put lines 98-247 in the discussion ( in a more concise manner).
Line 200-201. Was the test using 30 protein-ligand complexes performed in this study or from the literature? If it were from this study, please provide the result (could be in the supplementary material). If it were from the literature, please provide a sufficient argument that that test is relevant to support this study.
line 210. The subheader title (2.2.3) may be inaccurate
Line 259-260. Please describe how the LE value was obtained (in the material and method or somewhere in the introduction when the LE notation is introduced). Was it something generated by the program or calculated separately?
Line 266. The result also depends on the program used. It was not explicitly mentioned that the input parameters in the programs were of equal value.
Line 273-274. Please explain what “informative” mean. How does the author qualify the claim that more lead compound is better? If the screening that resulted in a small number of lead compounds may exclude compounds of actual potency, why is it not possible that having a large number means the inclusion of non-active compounds (which means that the screening efficiency is low, thus losing the point of improving the screening quality)?
Line 285. Please explain what other studied programs mean. Is it Glides?
Line 298-299. The ratio of the number of compounds in these three groups remained constant. are they compounds with similar structures? Perhaps from this point, the author could already form a preliminary prediction of which structure group the biologically active compound belongs to.
Line 326-328. Please clarify the sentence. It seems like three sentences were being merged.
Line 375-376. Please clarify the sentence.
Line 380-382. Regarding the task of Glide HTVS, is it part of the strategy or the primary function of the program?
Line 322-474. The results appear to suggest that there were no reliable rankings for all the programs used. If there was, please make it more obvious. On the use of the Glide family (HTVS, SP, XP), is it a convergence approach (starting from the less stringent to the strict one)? Or eventually, the output from those three can be combined to recommend the most potential compounds (e.g. The hotspot compounds, which are those ranked in the top 5 in each).
Line 515-536. The conclusion should be much shorter. It should only be mentioned what are the outcomes of the study (and suggestions or recommendations). At present, the conclusion part is more of a summary. The appropriate conclusion of the manuscript would be in these 2 sentences: (1) “results and threshold .... target protein” (line 526-527), and (2) “Thus, the rational ..... drug development” (line 533-536).
Other general comments/suggestions.
-
Many sentences are too complex. Please write in a more simple and more concise manner. There are sentences with 2-3 subjects, hence it is difficult to understand which information belongs to whom.
-
Instead of presenting as per the docking program, it is better to present the result as per protein. For example, for AchE, the result of Autodock was AAA, Vina was BBB, Glide 1 was CCC, and so on. This way, it is easier to compare the performance of the programs and at the end, a trend can be drawn from the output of the program for each protein. It is more difficult to compare the output from each protein as per the program since each protein is unique, has different ligands, and has structures of different resolutions.
Author Response
First of all, we would like to thank the reviewer for their useful recommendations and suggestions. The detailed answers to the reviewer are listed below:
- The application would be more appealing for use if there is an input for prior knowledge of the protein. For example, the protein X (the target) is known to have affinities towards K, L, M, and N types of compounds. An example of this is well provided in 2.1.2. This strategy should be made more profound. Answer: The detailed study of the relationship between compound structure, type of protein, and affinity of the target to the exact type of compound was not the goal of this work. The focus of this work is the study of the dependence of the predictive power of individual docking programs for different types of proteins and for different types of compounds.
- It should be mentioned whether this would work to evaluate compounds that have been known or also for new compounds. Answer: The main goal of the current manuscript is the attempt to find out the potential threshold values of binding energy and ligand efficiency that would allow for a more reliable ranking of compounds during HTVS and molecular docking. The prediction of new compounds is not the topic of this work.
- Considering that the main focus of the manuscript is screening and/or evaluation of potential molecules for drug development, it is surprising that Lipinski's rule is not mentioned at all because drug development would involve ADME at some points. Answer: In the context of the current manuscript, Lipinski’s rule was not taken into account. This rule and its modification usually are used in the compound library preparation step (in order to decrease its size) or as an additional condition for the identification and selection of the best drug candidate for further biological evaluation. Moreover, the study of the relationship between the binding energy and compliance of the drug candidate with Lipinski's rule may be the subject of a separate publication.
- The molecular docking study in the manuscript employed several proteins. The protein structural model used are varying in resolution quality: AChE is 2.40Å, BACE1 is 1.35Å, GSK-3β is 2.40Å, TS3 is 3.14Å (5I6X) and 3.24Å (5I73), and TrkA 2.75Å. Protein structures themselves are models, thus there is an indigenous error (which can be checked through their R factors) and the resolution of the structure affects docking precision, especially when comparing the binding energies. Answer: In the context of this work, the impact of the crystal structure (model) used on the molecular docking results was not considered. Also, the assumptions and simplifications that are used in the molecular docking methodology can have a greater impact on the docking results than the errors in the used 3D model or crystal structure of the target. It should be also mentioned that the resolution of most of the crystal structures used in this work is appropriate for docking purposes (lower than 3Å). Only for TS3 (PDB IDs: 5I6X and 5I73), this value is exceeded by 0.14 and 0.24 respectively since there was no available lower-resolution human SERT structure in Protein Data Bank.
Detailed comments
Line 25-90. Paragraph 1 is very long, it can be divided into several paragraphs. Answer: We agree with the reviewer that the introduction may seem too long. However, this is most likely due to both the number of programs used and the number of studied protein targets. The main purpose of the introduction is to give the reader enough necessary information about the subject of research. In our opinion, at the moment the introduction meets this requirement. The division into separate paragraphs may be inconvenient and confusing for the reader, and may also increase the length of the introduction, which is undesirable.
Line 66-67. Please make clear what basic parameters were modified. Answer: In their work, Cross et al. studied the effects of changing a large number of different baseline parameters in all programs studied. In our opinion, the enumeration and detailed description of this does not correspond to the purpose of the introduction. Moreover, such information can significantly increase the size of the introduction, which is undesirable.
Line 69-71. It is confusing. There are 10 docking programs tested, and the same docking tool (does it mean one of the programs?) gives different outputs for the different protein families. Please clarify. Answer: Wang et al. reported that the predictive power of each of the ten studied docking programs was quite different for different protein families. In the revised version of the manuscript, this sentence is corrected.
Line 76. Using quantum mechanics is said to be more accurate. However, usually, quantum mechanics is calculated in a vacuum field, which may not be appropriate to calculate the energy. Please explain. Answer: In this sentence, we do not claim that the quantum mechanics calculation of the binding energy is more accurate. The sentence mentioned that QM provides a more accurate calculation of the non-covalent interactions which cannot be achieved with molecular docking. The QM calculation of non-covalent interactions is very useful as a tool to test and improve molecular mechanics force fields and to model the forces involved in biomolecular binding and folding.
Line 98-160. (2.1. Selection of biological targets and compounds library) and lines 161-247. (2.2. Molecular docking software). These parts are better distributed into the introduction and material and methods because it is hardly present and discuss the results of the experiment. Another option is: separate the result and discussion into two sections and put lines 98-247 in the discussion (in a more concise manner). Answer: In the revised manuscript “Section 2.1.2” is moved to the materials and methods part (In the current version, it is “Section 3.1.2”). The selection of biological targets as well as molecular docking programs' historical and methodological review is included in the introduction part (as different sections).
Line 200-201. Was the test using 30 protein-ligand complexes performed in this study or from the literature? If it were from this study, please provide the result (could be in the supplementary material). If it were from the literature, please provide a sufficient argument that that test is relevant to support this study. Answer: This test was carried out during the development of the AutoDock and AutoDock Vina. Now, a reference is added (p.6, line 204). The main purpose of this mention is historical and methodological information about the programs used.
line 210. The subheader title (2.2.3) may be inaccurate. Answer: The subheader title is corrected.
Line 259-260. Please describe how the LE value was obtained (in the material and method or somewhere in the introduction when the LE notation is introduced). Was it something generated by the program or calculated separately? Answer: The information about ligand efficiency calculations is currently added to the materials and method section (p. 16, lines 533-536).
Line 266. The result also depends on the program used. It was not explicitly mentioned that the input parameters in the programs were of equal value. Answer: This sentence contains this information: “However, it is noticeable that the composition of the final set of compounds, as well as its size, depends not only on the type of target protein but also on the software used” (p. 7, lines 264 – 266).
Line 273-274. Please explain what “informative” mean. How does the author qualify the claim that more lead compound is better? If the screening that resulted in a small number of lead compounds may exclude compounds of actual potency, why is it not possible that having a large number means the inclusion of non-active compounds (which means that the screening efficiency is low, thus losing the point of improving the screening quality)? Answer: In the current version of the manuscript, the word “informative” is replaced by “reliable”. In the context of this work, the claim that “more lead compound is better” is justified. If the used threshold value allows us to identify all 50 lead compounds of the benchmark dataset it means that this approach can be used the improving the HTVS results. If, on the other hand, the selected threshold value does not allow us to identify any lead compound from the benchmark data set, then we cannot say that this is efficient and informative for further study.
Line 285. Please explain what other studied programs mean. Is it Glides? Answer: This means Glide docking programs and AutoDock 4.2.6. This information is added to the manuscript (p. 7, line 286).
Line 298-299. The ratio of the number of compounds in these three groups remained constant. are they compounds with similar structures? Perhaps from this point, the author could already form a preliminary prediction of which structure group the biologically active compound belongs to. Answer: The used benchmark datasets for each target protein were prepared as follows: (I) newer experimental data were preferable, (II) where possible, the same or similar experimental protocol for activity determination was used, and (III) compounds with diverse structures were selected (pp. 15-16, lines 489-504). Undoubtedly, such preparation does not exclude the possibility that some compounds may be structurally similar. However, a detailed study of the structure–binding energy (or ligand efficiency) relationship was not the aim of this study.
Line 326-328. Please clarify the sentence. It seems like three sentences were being merged. Answer: This is one sentence and cannot be divided into 3 sentences without losing the original meaning.
Line 375-376. Please clarify the sentence. Answer: In the revised manuscript, this sentence is corrected.
Line 380-382. Regarding the task of Glide HTVS, is it part of the strategy or the primary function of the program? Answer: This is the primary function of the program.
Line 322-474. The results appear to suggest that there were no reliable rankings for all the programs used. If there was, please make it more obvious. On the use of the Glide family (HTVS, SP, XP), is it a convergence approach (starting from the less stringent to the strict one)? Or eventually, the output from those three can be combined to recommend the most potential compounds (e.g. The hotspot compounds, which are those ranked in the top 5 in each). Answer: Indeed, our results showed that none of the programs allows us to reliably rank compounds according to their biological activity. This information is mentioned on p. 10, lines 327 - 328 and 339 - 341; on p. 11, lines 356 - 359; on p. 12, lines 377 - 379 and 410 - 412; on p. 14, lines 461 - 463; on p. 15, lines 475 - 477. Such formulations were chosen by us in order to avoid presenting the results in a negative way.
The strategy of starting from a less stringent score evaluation to the strict one is the original Glide VSW method. In this procedure (Glide VSW), all three precision levels (HTVS, SP, and XP) are used in order one after the other. The number of compounds, as well as the compound selection criterion (for example, only best scoring pose) for the next step (next precision level), can be selected by the user depending on the task. In this work, we have concentrated on testing the predictive power of each of the three precision levels when used alone.
Line 515-536. The conclusion should be much shorter. It should only be mentioned what are the outcomes of the study (and suggestions or recommendations). At present, the conclusion part is more of a summary. The appropriate conclusion of the manuscript would be in these 2 sentences: (1) “results and threshold .... target protein” (line 526-527), and (2) “Thus, the rational ..... drug development” (line 533-536). Answer: We thank the reviewer for this suggestion. The main result of this work is that each target protein threshold value of the binding energy or LE should be selected depending not only on the type and structure of the protein but also on the software used. In the current version of the manuscript, these outcomes as well as recommendations are indicated. Enumeration of each of the suggested binding energy thresholds for each of the five targets studied in the case of all programs studied can greatly complicate and increase the conclusions, which may be undesirable.
Other general comments/suggestions.
- Many sentences are too complex. Please write in a more simple and more concise manner. There are sentences with 2-3 subjects, hence it is difficult to understand which information belongs to whom. Answer: We have gone once more through the English checking and revised the long sentences. However, in some cases, it is not possible to shorten a sentence without losing the original meaning. It should be also mentioned that we are going to use the MDPI English checking service in order to improve the quality of the manuscript.
- Instead of presenting as per the docking program, it is better to present the result as per protein. For example, for AchE, the result of Autodock was AAA, Vina was BBB, Glide 1 was CCC, and so on. This way, it is easier to compare the performance of the programs and at the end, a trend can be drawn from the output of the program for each protein. It is more difficult to compare the output from each protein as per the program since each protein is unique, has different ligands, and has structures of different resolutions. Answer: During the writing of the work, we tried to present the results in the order suggested by the reviewer. However, this order of presentation turned out to be not entirely convenient and logical, so it was decided to stop at the current version of the presentation of the results.
Round 2
Reviewer 1 Report
Recommendation: This paper maybe be accepted based on responses from other reviewer ( 3)